# Association of Zinc Deficiency with Development of CVD Events in Patients with CKD

**DOI:** 10.3390/nu13051680

**Published:** 2021-05-15

**Authors:** Shinya Nakatani, Katsuhito Mori, Tetsuo Shoji, Masanori Emoto

**Affiliations:** 1Department of Metabolism, Endocrinology and Molecular Medicine, Osaka City University Graduate School of Medicine, 1-4-3 Asahi-machi, Abeno-ku, Osaka 545-8585, Japan; m2026719@med.osaka-cu.ac.jp (S.N.); memoto@med.osaka-cu.ac.jp (M.E.); 2Department of Nephrology, Osaka City University Graduate School of Medicine, 1-4-3 Asahi-machi, Abeno-ku, Osaka 545-8585, Japan; 3Department of Vascular Medicine, Osaka City University Graduate School of Medicine, 1-4-3 Asahi-machi, Abeno-ku, Osaka 545-8585, Japan; t-shoji@med.osaka-cu.ac.jp; 4Vascular Science Center for Translational Research, Osaka City University Graduate School of Medicine, 1-4-3 Asahi-machi, Abeno-ku, Osaka 545-8585, Japan

**Keywords:** zinc, hemodialysis, chronic kidney disease, cardiovascular disease

## Abstract

Deficiency of the micronutrient zinc is common in patients with chronic kidney disease (CKD). The aim of this review is to summarize evidence presented in literature for consolidation of current knowledge regarding zinc status in CKD patients, including those undergoing hemodialysis. Zinc deficiency is known to be associated with various risk factors for cardiovascular disease (CVD), such as increased blood pressure, dyslipidemia, type 2 diabetes mellitus, inflammation, and oxidative stress. Zinc may protect against phosphate-induced arterial calcification by suppressing activation of nuclear factor kappa light chain enhancer of activated B. Serum zinc levels have been shown to be positively correlated with T_50_ (shorter T_50_ indicates higher calcification propensity) in patients with type 2 diabetes mellitus as well as those with CKD. Additionally, higher intake of dietary zinc was associated with a lower risk of severe abdominal aortic calcification. In hemodialysis patients, the beneficial effects of zinc supplementation in relation to serum zinc and oxidative stress levels was demonstrated in a meta-analysis of 15 randomized controlled trials. Thus, evidence presented supports important roles of zinc regarding antioxidative stress and suppression of calcification and indicates that zinc intake/supplementation may help to ameliorate CVD risk factors in CKD patients.

## 1. Introduction

The micronutrient zinc is an essential trace element and the second most abundant divalent cation in the body (2–4 g), with approximately 57% existing in skeletal muscle and 29% in bone [1]. Inadequate intake, decreased absorption, and/or increased loss of zinc can result in a deficiency. Zinc deficiency is common throughout the world and affects over two billion people [2]. In patients with chronic kidney disease (CKD), several previous studies have demonstrated lower blood zinc levels, with the prevalence of zinc deficiency ranging from 40% to 78% in those undergoing hemodialysis [3,4].

Zinc plays important roles in various biochemical pathways, including as a cofactor with >300 enzymes. Additionally, zinc is involved in structural integrity maintenance, basic cellular functions such as proliferation, DNA and RNA synthesis, control of the expression of several genes, and regulation of the immune functions of many types of cells [5,6]. In previous reports, zinc deficiency has been shown to be associated with growth disturbance [7,8], taste impairment [9], anorexia and loss of appetite [10], dermatitis [11], delayed wound healing [12], and infection [13]. Zinc is also essential in an active site of superoxide dismutase (SOD), an important antioxidant enzyme that catalyzes the dismutation of superoxide [14]. Furthermore, treatment with zinc has been found to attenuate reactive oxygen species (ROS) production [15].

In a recent in vitro study, zinc attenuated phosphate-induced osteochondrogenic phenotypic switch of vasculature smooth muscle cells (VSMCs) leading to development of vascular calcification [16]. Additionally, an in vivo study demonstrated that zinc can protect against phosphate-induced arterial calcification by inducing production of a zinc-finger protein, tumor necrosis factor (TNF)-a-induced protein 3 (TNFAIP3), and suppressing activation of nuclear factor kappa-light-chain-enhancer of activated B (NF-k B) [17].

Cardiovascular disease (CVD) is the leading cause of morbidity and mortality throughout the world [18,19,20]. Furthermore, vascular calcification is a striking feature of chronic inflammatory diseases including CKD and has been shown to be associated with increased risk of CVD events [18,20]. An in vitro test (T_50_-test) for determination of serum calcification propensity has been developed [21], as that has been shown to be a novel surrogate marker of CVD events [22]. A shorter T_50_ means a higher calcification propensity [21]. In other studies, serum zinc levels were found to be positively correlated with T_50_ in patients with type 2 diabetes mellitus [23] and CKD [17], and higher intake of dietary zinc was independently associated with lower risk of severe abdominal aortic calcification (AAC) in noninstitutionalized adults in the United States [24]. Indeed, several cohort studies have shown that low zinc intake is associated with cardiovascular mortality [25,26].

Given the importance of zinc regarding both attenuating antioxidative stress and suppressing calcification, it is not surprising that accumulated evidence suggests an association of zinc deficiency with the development of CVD events in patients with CKD, especially those undergoing hemodialysis. The aim of this review is to summarize results presented in literature so as to consolidate current knowledge regarding zinc status, including zinc status in patients affected by CKD and receiving hemodialysis treatments.

## 2. Zinc and Nutrition

The daily requirement of zinc for adults throughout the world ranges from 7 to 11 mg [27]. In Japan, the recommended dietary intake (RDI) of zinc for adults is 9–10 mg/day for males and 7–8 mg/day for females [28], although intake is insufficient in 60–70% of both genders aged over 20 years [29]. Additionally, zinc intake was found to be lower in patients with advanced stage CKD as compared with those without advanced stage CKD who participated in the National Health and Nutrition Examination Survey conducted in the United States [24]. 

Meats as well as oysters and scallops contain abundant zinc [30], while some cereals and Japanese foods, including tofu, rice, and fermented soybeans (*natto*), also have large amounts [29]. Generally, red meat contains more zinc than white meat and fish [30]. It is also known that leaf vegetables and fruits have low zinc concentrations, along with high water and potassium content [31]; thus, patients undergoing hemodialysis must be careful regarding their overconsumption so as to avoid hyperkalemia and volume overload.

It is important to recognize that the content of zinc in food (nutrients) is not necessarily identical with its availability [30,32]. For example, the availability of zinc present in peas, lentils, and beans is limited because it is released during milling [30]. In addition, food components, such as phytate [27], casein [27,31], and fiber, as well as higher calcium concentrations are known to impair zinc absorption [33]. Phytate forms insoluble complexes with zinc in the intestines [27,34], which then hinder its absorption and bioavailability; consequently, zinc absorption is reduced in intestinal cells [35]. It is very likely that the absorption-inhibiting effects of fiber and higher calcium concentrations are also due to the phytate they contain [33]. Methods used for processing, such as grinding, soaking, germination, malting, and/or fermentation, can reduce the phytate content of foods and thus its protective effect in regard to absorption of zinc [31]. Furthermore, various organic acids including citric acid (citrus fruits) and lactic acid (sour milk) as well as fruit acid bind to zinc and increase its absorption [31].

## 3. Zinc Deficiency in CKD

Reduced levels of zinc in serum or plasma of patients with CKD have been demonstrated (Table 1). Several factors may contribute to and explain zinc deficiency seen in association with CKD. Some studies have demonstrated a negative zinc balance in CKD patients [36,37]. This may be due to decreased intestinal absorption, decreased food intake, uremic toxicity, bioavailability, and/or increased loss, such as through the face, urine, or hemodialysis. It is also important to note that many CKD patients are elderly and using multiple medications that can affect taste sensation and increase zinc deficiency [38]. In a recent study by Chen et al., 5/6 nephrectomized rats as well as phenylhydrazine-induced anemic mice were found to have zinc redistributed to bone marrow from bone and plasma, causing the zinc level in plasma to decrease, which produced reticulocytes [39]. Their novel findings of pools of zinc in plasma and bone redistributed to bone marrow in the majority of those nephrectomized rats indicate the mechanism of zinc deficiency in CKD. 

### 3.1. Urinary Zinc Excretion in CKD

Due to the fact that zinc is bound to proteins in plasma [52], it is generally believed that glomerular filtration of zinc and consecutive urinary zinc excretion are limited [1]. However, Damianaki et al. recently found that urinary zinc excretion was significantly higher in CKD patients (612.4 ± 425.9 μg/24 h) (*n* = 108) as compared with non-CKD patients with preserved kidney function (479.2 ± 293.0 μg/24 h) (*n* = 81) (*p* = 0.02) [42]. That study also noted that zinc fractional excretion was stable in the early stage of CKD, then a sudden and strong increase was seen in stage 3 patients, while that was correlated negatively and linearly with estimated glomerular filtration rate (eGFR). Although the mechanism related to increased fractional excretion of zinc in CKD patients remains unclear, increased urinary zinc excretion has been shown to be linked to tubular dysfunction in patients with cancer [53], type 1 diabetes mellitus [54], and type 2 diabetes mellitus [55], possibly due to impaired tubular activity. In the study presented by Damianaki et al., zinc fractional excretion was correlated negatively with 24 h urinary uromodulin excretion (*r* = 0.29; *p* < 0.01) [42]. Uromodulin, a protein produced by tubular cells of the ascending loops of Henle, was recently identified as a marker of tubular mass and function in the general population [56]. The negative correlation between urinary uromodulin excretion and fractional excretion of zinc suggests that urinary zinc loss may be linked to the low number of functional tubules associated with CKD.

### 3.2. Taste Change Associated with CKD

Many CKD patients are elderly and using multiple medications that can affect taste sensations and increase zinc deficiency [38]. Taste change has been reported by 40–60% of pre-dialysis CKD [57] and hemodialysis [58,59] patients, with taste changes of “bland” and “bitter” found to be associated with upper gastrointestinal symptoms, including nausea, vomiting, anorexia, and malnutrition [59].

In addition, taste change is one of the major symptoms of zinc deficiency [29]. There are roughly 7000 taste buds, peripheral receptors of taste, present in the oral cavity, pharynx, and larynx, with a particularly high concentration in lingual papillae on the tongue surface [29]. In findings obtained with an in vivo zinc-deficient model, microstructural abnormalities including microvilli rupture and vacuolation were shown in taste cells [60]. Taste cell differentiation from basal cells is also impaired when zinc deficiency is present [61], and an in vitro study demonstrated that reduced expression of bitter taste receptors was the result of that deficiency [62]. Furthermore, several reports have noted recovery of taste change by treatment with various forms of zinc, such as zinc gluconate [63,64], zinc picolinate [65], and polaprezinc [66]. Additional studies are needed to investigate whether recovery of taste change by use of these medications has effects on clinical hard endpoints such as CVD events or mortality.

### 3.3. Albumin and Zinc in CKD

Zinc is actively absorbed throughout the small intestine, and in circulation, zinc is present predominantly as being bound to proteins such as albumin, α-macroglobulin, and transferrin, with approximately 60–80% of zinc in serum bound to albumin [67,68]. Patients undergoing hemodialysis often show low serum albumin along with chronic malnutrition. Because albumin is the primary carrier protein for circulating zinc [69], hypoalbuminemia should be considered as another confounding factor in interpreting plasma zinc concentration.

### 3.4. Other Factors of Zinc Deficiency in CKD

Several hypotheses related to low zinc levels in CKD patients have been proposed, such as low dietary intake due to protein restriction and decreased gastrointestinal absorption of dietary zinc due to impairment of formation of 1.25-dihydroxycholecalciferol or drug interactions [70,71]. Indeed, zinc intake has been shown to be lower in patients with advanced stage CKD [24]. In addition, phosphate binding decreases zinc absorption [72]. Physicians should pay careful attention to possible effects on plasma zinc levels when they start protein restriction and use phosphate binder medication in the management of patients with CKD.

## 4. Vascular Calcification and CVD in CKD

One of the most characteristic features of vascular change seen in dialysis patients is vascular calcification, which is associated with several types of target organ damage, including stroke, ischemic heart disease, and peripheral arterial disease. Gorriz et al. demonstrated that vascular calcification is a predictor of cardiovascular death in patients with varying CKD stages before starting dialysis [19].

CVD events are well known to increase as kidney function declines. The largest population-based study, conducted by Go et al. with 1,120,295 adult subjects, revealed that the adjusted hazard ratio (HR) for CVD events was inversely associated with eGFR, with an HR of 1.4 for subjects with an eGFR of 45–59 mL/min/1.73 m^2^, 2.0 for those with an eGFR of 30–44 mL/min/1.73 m^2^, 2.8 for those with an eGFR of 15–29 mL/min/per1.73 m^2^, and 3.4 for those with an eGFR <15 mL/min/1.73 m^2^ [73]. Additionally, adjusted risk of hospitalization and mortality followed a similar pattern. A collaborative meta-analysis of 10 cohorts with a total of 266,975 patients performed by van der Velde et al. revealed a similar trend showing a relationship of CVD mortality with eGFR [74].

## 5. Vascular Calcification and Vascular Smooth Muscle Cells

The pathophysiology of vascular calcification in CKD patients involves several factors, including changes related to oxidative stress, inflammation, imbalance between calcification promoters and inhibitors, and extracellular matrix metabolism, as well as calcium and phosphate metabolism imbalances [75,76].

### 5.1. Vascular Smooth Muscle Cells

Excess CVD morbidity and mortality in CKD patients might be explained by redistribution and/or overload of calcium and phosphorus. The primary mechanism of vascular calcification is considered to be related to ectopic deposition of hydroxyapatite [77] induced by an increase in calcium–phosphorus product (Ca × P) in serum [78]. VSMCs play a pivotal tool for investigation of vascular calcification [79]. Previous studies have reported transdifferentiation of VSMCs into osteoblast-like cells [80,81]. Bone morphogenetic protein-2 (BMP-2), oxidized lipids, and inflammation are known to accelerate vascular calcification [82], whereas matrix Gla protein (MGP), osteoprotegerin, and osteopontin act on the vascular wall as calcification inhibitors [83]. A recent work focused on runt-related transcription factor 2 (Runx2), an essential transcriptional factor for osteogenesis, and reported that Runx2 appears to be involved in repression of the primary VSMC phenotype, in addition to acceleration of the osteogenic phenotype [84]. Phenotypical transdifferentiation of VSMCs is regulated by complex signaling pathways [85] that are linked to generalized inflammation and dependent, at least in part, on the transcription factor NF-kB, which has emerged as a key regulator of vascular calcification [86].

### 5.2. Zinc Inhibits Phosphate-Induced VSMC Calcification

Some recently presented studies reported that zinc plays an important role in inhibition of calcification using VSMCs. Zinc sulfate blunted phosphate-induced calcification and decreased messenger RNA expression by osteogenic markers including TNFAIP3 expression, which subsequently was shown to inhibit NF-kB activation and osteo-/chondrogenic reprograming, resulting in suppression of phosphate-induced calcification of VSMCs [17] (Figure 1).

Hypoxia-inducible factor (HIF) stabilizers, also known as HIF prolyl hydroxylase inhibitors (PHI), are promising candidates for treatment of CKD-associated anemia as they increase erythropoietin synthesis [87,88]. Moreover, recent findings suggest that HIFs also play a pivotal role in vascular calcification [89,90]. Indeed, FG4592, an orally bioavailable PHI, promoted phosphate-induced loss of smooth muscle cell markers (ACTA-2, MYH11, SM22a) and enhanced osteochondrogenic gene expression (Msx-2, BMP-2, Sp7), thus triggering an osteochondrogenic phenotypic switch by VSMCs [16]. These effects of PHI occurred in parallel with increased pyruvate dehydrogenase kinase 4 (PDK4) expression. Additionally, zinc was shown to inhibit an osteochondrogenic phenotypic switch of VSMCs, reflected by lowered phosphate uptake, which resulted in decreased expressions of Msx-2, BMP-2, and Sp7, as well as loss of smooth muscle cell specific markers. That previous study [16] also found that zinc preserved the phosphorylation state of Runx2, decreased PDK4 level, and restored cell viability, suggesting that it inhibits PHI aggravated by VSMC calcification induced by a high phosphate level (Figure 1). Clinical trials are needed to examine whether zinc supplementation attenuates aortic calcification and CVD events in CKD patients with high phosphate overload or treated with PHI.

## 6. Zinc and Calcification Propensity in Serum

### 6.1. Serum Calcification Propensity (T_50_)

In serum, precipitation of supersaturated calcium and phosphate is prevented by formation of amorphous primary calciprotein particles [91,92]. Primary calciprotein particles spontaneously convert into secondary calciprotein particles that contain crystalline hydroxyapatite [91,92]. The propensity for transformation to secondary calciprotein particles can be assessed by determining the time required to transform into secondary calciprotein particles, termed serum calcification propensity or T_50_, and an in vitro test (T_50_-test) has been developed for determination of serum calcification propensity [21]. This assay determines the time required for primary calciprotein particles to transform into secondary calciprotein particles in the presence of supersaturating doses of calcium and phosphate, resulting in increased turbidity of the samples. Serum T_50_ can be examined by laser light scatter in turbid samples using nephelometry and a shorter T_50_ time means a higher calcification propensity. Studies have shown that lower T_50_ predicts vascular stiffness progression and all-cause mortality in patients with stage 3 or 4 CKD [22], as well as all-cause mortality and cardiovascular composite endpoint in hemodialysis patients [93]. A lower T_50_ level was also shown to predict cardiovascular and all-cause mortality in renal transplant recipients [94,95]. While it remains unclear how well in vitro T_50_ assay results represent the mineralization process in vivo, the mineralization process has been shown to be associated with arterial calcification, arterial stiffness, cardiovascular outcomes, and mortality in at least 18 observational and 11 interventional studies [96]. Thus, T_50_ may be useful as a surrogate marker of calcification stress which is closely associated with CVD risk in CKD.

### 6.2. Association of Zinc and Serum Calcification Propensity

A cross-sectional study of 132 type 2 diabetes mellitus patients with various levels of kidney function showed a weak but positive correlation of serum zinc with T_50_ [23], while another study that included healthy subjects and patients with CKD also reported a positive correlation [17]. Furthermore, in vitro experiments demonstrated that addition of a physiological concentration of exogenous zinc chloride significantly increased serum T_50_. Together, these findings indicate that serum zinc is an independent factor with a potential role in suppression of calcification propensity in serum [23].

## 7. Zinc and Vascular Change

### 7.1. Zinc and Abdominal Aortic Calcification

AAC is common in CKD cases and is known to be an independent predictor of cardiovascular mortality in both the general population and CKD patients [97,98]. Recently, Chen et al. showed that high dietary zinc intake was independently associated with lower risk of severe AAC in non-institutionalized adults in the United States (*n* = 2535) [24]. In that study, 18.1% of the subjects were CKD patients, and higher zinc intake in those was associated with reduced risk of severe AAC after adjustment for age, gender, and ethnicity, while no association between zinc intake and AAC was found in the fully adjusted model. However, that lack of association might have been due to the low statistical power due to the small sample size.

### 7.2. Zinc and Carotid Intima-Media Thickness

Carotid intima-media thickness (CIMT) is a valuable marker of subclinical atherosclerosis [99]. In a study that included middle-aged and elderly subjects, the low zinc intake group showed a greater CIMT than the high zinc intake group [100]. A reduced serum zinc level is also associated with increased CIMT in patients receiving hemodialysis [101]. To clarify the effects of zinc intake and/or supplementation for AAC and/or CIMT, further studies focused on advanced CKD stage are necessary.

## 8. Zinc Deficiency and Risk Factors for CVD

Zinc deficiency is also associated with various risk factors related to CVD events, such as high blood pressure, dyslipidemia, type 2 diabetes mellitus, inflammation, and oxidative stress (Figure 2).

### 8.1. Zinc Deficiency and Blood Pressure

Zinc is also known to be involved in arterial pressure regulation. In salt-sensitive hypertensive model rats, plasma zinc levels were found to be reduced [102]. Additionally, in spontaneous hypertension-prone rats, dietary zinc restriction exacerbated systolic blood pressure [102], whereas zinc supplementation attenuated blood pressure response [103]. Of interest, Williams et al. revealed a possible mechanism of zinc related to blood pressure regulation, as an Na^+^-Cl^−^ cotransporter (NCC) was found to be a zinc-regulated transporter upregulated in mice with zinc deficiency, and NCC upregulation contributed to increased blood pressure by stimulating renal sodium reabsorption [104]. In human studies, it has been reported that populations with low dietary zinc intake have a high prevalence of hypertension, and a possible inverse correlation between zinc levels and blood pressure has also been noted [105,106].

### 8.2. Zinc Deficiency and Dyslipidemia

Animal studies have demonstrated profound effects of zinc deficiency on the cell structure of the aorta, as well as fatty acids and carbohydrate metabolism, which are disadvantageous for maintaining vascular health [107]. In low-density lipoprotein (LDL) receptor knock-out mice, acute zinc deficiency elicited changes in key transcription factors and adhesion molecules found to be pro-atherogenic [108]. In addition, zinc is a cofactor for desaturases and elongases involved in endogenous fatty acid synthesis [109,110]; thus, an alteration in its plasma level may influence the activities of these enzymes and consequently regulation of fatty acid metabolism.

A systematic review and meta-analysis of 24 studies that included 14,515 subjects showed favorable effects of zinc supplementation on lipid parameters [111]. It was found that zinc supplementation (average 39.3 mg/day) achieved a significant reduction in LDL cholesterol (−4.78 mg/dL) and total cholesterol (−10.72 mg/dL), as well as triglycerides (−8.73 mg/dL). Hypercholesterolemia and hypertriglyceridemia have been reported in previous studies that used a zinc-deficient diet, which might induce CVD events and insulin resistance in CKD patients [112,113], while others have suggested that zinc supplementation improves blood lipid metabolism in hemodialysis patients [114,115]. More evidence is needed to better understand the effects of zinc supplementation on the lipid profile of patients undergoing hemodialysis.

### 8.3. Zinc Deficiency and Type 2 Diabetes

Zinc deficiency is also an important risk factor in regard to type 2 diabetes mellitus [116,117]. Although the mechanism by which zinc may have an impact on risk of type 2 diabetes mellitus development has not been completely elucidated, zinc is known to participate in adequate insulin synthesis, storage, crystallization, and secretion in pancreatic β-cells, as well as be involved in the action and translocation of insulin into cells [118,119]. In addition, zinc apparently has a role in insulin sensitivity via activation of the phosphoinositol-3-kinase/protein kinase B cascade [119]. Due to its insulin–mimetic action, zinc also stimulates glucose uptake in insulin-dependent tissues [120].

Regarding zinc intake, a systematic review and meta-analysis of prior cohort showed that a moderately high dietary zinc intake in relation to the dietary recommended intake values was associated with a lower risk of type 2 diabetes mellitus by 13%, and it was associated with 41% lower risk in subjects living in rural areas [121]. A randomized controlled trial (RCT) also showed that zinc supplementation improved glucose metabolism and insulin sensitivity in diabetic patients [122]. Nevertheless, additional evidence is needed to clarify the effects of zinc intake and/or supplementation on glucose metabolism, as well as the association between blood zinc status and glycemic status in patients undergoing hemodialysis.

### 8.4. Zinc Deficiency and Inflammation

Inflammation is another major risk factor that contributes to the pathological process of CVD. Zinc is well known to be essential for normal functions of the immune system in both innate and adaptive immunity responses to pathogens or tissue damage [123]. NF-κB, a transcription factor and key modulator in inflammatory response pathways [124], has an ability to enter the nucleus and induce expression of targeted genes. NF-κB-regulated genes include a variety of inflammatory cytokines, such as interleukin-1 (IL-1), IL-6, TNF-α, lymphotoxin, and interferon-γ (IFN-γ) [123]. Dietary zinc deficiency and intracellular zinc deprivation have been shown to result in increased activation of NF-κB, as well as inflammatory cytokine expression in cultured cells and animal models regulated by NF-κB [125,126]. Supplementation with zinc might also suppress NF-κB activation and NF-κB-regulated inflammatory cytokine release [127].

Furthermore, in a calcification model with klotho-hypomorphic, subtotal nephrectomy, and cholecalciferol overload mice, zinc sulfate supplementation was found to increase aortic expression of the zinc finger proteinTNFAIP3, which is a suppressor of the NF-kB transcription factor pathway and which is reused when NF-κB is suppressed [17]. Supplementation with zinc may suppress NF-κB activation and NF-κB-regulated inflammatory cytokine release, as well as aortic calcification, likely by inhibiting phosphorylation and degradation of IκB and increasing TNFAIP3.

Mousavi et al. (2018) showed a significant reduction in circulating C-reactive protein (CRP) level after zinc supplementation and concluded that such supplementation might have a beneficial effect on serum CRP, especially at a dose of 50 mg/day, in adult kidney failure patients [128].

### 8.5. Zinc Deficiency and Oxidative Stress

Oxidative stress is also a key risk factor that contributes to the development and progression of CVD [129]. Nuclear factor erythroid 2-related factor 2 (Nrf2) is a master transcriptional regulator of genes related to redox status and antioxidant effects [130]. Zinc has been shown to be involved in modulation of Nrf2 [14]. In endothelial cells, zinc positively regulates glutamate cysteine ligase expression by activating and promoting translocation of the transcription factor Nrf2 to the nucleus [131], while it also activates the antioxidant responsive element–Nrf2 pathway in epithelial cells [132]. In human renal tubular cells under diabetic conditions, treatment with zinc significantly increased nuclear expression of Nrf2 [133]. Human studies also showed that CKD patients have downregulation of Nrf2 mRNA expression [134]. In addition, bardoxolone methyl, a potent activator of Nrf2 was shown to increase eGFR in patients with type 2 diabetes mellitus and CKD stage 3 when administered over a 52-week period [135]. Thus, additional RCTs are needed to investigate whether zinc supplementation, such as with bardoxolone methyl, has protective effects on kidney function in CKD patients.

ROS production is increased by zinc deficiency in various cells, such as in mouse 3T3 cells, in human fibroblasts, and in neuronal and epithelial cells [136]. Zinc is also essential in the active site of SOD, an important antioxidant enzyme that catalyzes the dismutation of superoxide. In addition, zinc treatment of human peritoneal mesothelial cells was found to inhibit activation of the nucleotide-binding domain and leucine-rich repeat-containing family, pyrin domain-containing-3 (NLRP3) inflammasome, by attenuating ROS production [15]. Thus, zinc has an important role as an antioxidant agent [14,137]. Together, these results indicate that zinc supplementation may contribute to an increase in Nrf2 expression and SOD synthesis, as well as to an improvement in antioxidant defense, resulting in reduced CVD risk in CKD patients. Further studies are anticipated.

## 9. Zinc Levels and CVD Events

In a systematic review of prospective cohort studies regarding zinc status and CVD events, Chu et al. found that higher serum zinc level was associated with lower risk of a CVD event [138], especially in vulnerable populations, including individuals with type 2 diabetes mellitus [139] and patients referred for coronary angiography [140]. Only a few studies of hemodialysis patients have been conducted. Recently, Toda et al. prospectively investigated the association of zinc status and CVD events in 142 incident hemodialysis patients [141]. Although all patients undergoing hemodialysis are at risk for a CVD event, this longitudinal study (mean follow-up 2.5 years, 20 cases with CVD events) showed an insignificant association of a lower zinc level with a higher risk of CVD. The relatively small sample size is a limitation of that investigation. To clarify the relationship between serum zinc levels and cardiovascular events in patients with CKD and hemodialysis, studies with a greater number of cases will be necessary.

## 10. Zinc and CVD Mortality

### 10.1. Blood Zinc Level and CVD Mortality

We identified and summarized four cohort studies [25,129,131,132] that investigated the association between the level of zinc in blood and CVD mortality (Table 2). Results obtained in two of those showed increased risk in association with lower zinc level [25,140], while the other studies found no such association [142,143]. The subjects examined in the studies that showed increased risk with lower zinc level had increased CVD risk factors at the baseline, such as having been referred for coronary angiography. Thus, future studies that investigate blood zinc levels and CVD-related mortality in populations with increased risk for CVD, including CKD, hemodialysis, and type 2 diabetes mellitus, may provide additional important information. These four studies were conducted in Europe; thus, it will also be necessary to examine other subjects in other regions, as the sources of dietary zinc differ throughout the world.

### 10.2. Dietary Zinc Intake and CVD Mortality

General population studies have found that zinc intake is correlated with serum zinc level [144,145]. We investigated and summarized five cohort studies [25,26,146,147,148] that investigated the association between dietary zinc intake and CVD mortality (Table 3). Four studies showed a decreased risk of mortality in association with greater dietary zinc intake [25,26,135,137]. However, findings in the study showing no such association [147] could be explained, at least in part, by the relatively lower incidence of CVD mortality and adequate dietary zinc intake in those subjects. Higher dietary zinc intake likely has positive effects on CVD mortality. Nevertheless, results showing dietary zinc intake by CKD patients are few; thus, further observational and interventional studies that include those patients are needed.

To properly interpret previous studies of zinc intake, the various sources must be considered. In Japan, the main dietary source of zinc is rice [149], and higher dietary zinc intake was shown to be associated with lower mortality due to lower incidence of coronary heart disease [26]. In contrast, red meat is the main source of zinc in the United States [24], and a high level of consumption has been reported to be associated with vascular mortality [150], while increased iron load due to meat-derived heme iron intake has been discussed as a potential underlying mechanism [151,152].

## 11. Zinc and Progression of CKD

Damimanaki et al. assessed the relationship between baseline plasma zinc level and yearly kidney function decline in a cohort with 3-year follow-up data and found a significant association of a lower baseline zinc level with a large decline of kidney function [42]. Furthermore, the association remained statistically significant in multivariable models adjusted for age, gender, diabetes, and arterial hypertension, while it was no longer significant when baseline eGFR or proteinuria were introduced into the model [42]. Unfortunately, the number of subjects was relatively small (*n* = 108); thus, additional large scale longitudinal observational studies are necessary to clarify the association between blood zinc level and progression of CKD. Additionally, RCTs are needed to determine the effect of zinc supplementation on kidney function in pre-dialysis CKD patients.

## 12. Zinc Supplementation in Patients with CKD

Table 4 summarizes 17 RCTs [153,154,155] among which 15 trials were included in a previous systematic review and meta-analysis by Wang et al. [155]. Although the median intervention period was 60 days and the daily dose ~45 mg, zinc supplementation resulted in higher serum zinc, SOD, and dietary protein intake levels and lower levels of CRP and malondialdehyde [155]. In consideration of previously presented RCTs, zinc supplementation greater than 45 mg/day may be necessary to increase the serum zinc level in hemodialysis patients.

In children with CKD, adequate nutritional status is important for normal growth and development; thus, careful monitoring is essential [169]. The Chronic Kidney Disease in Children study revealed that 7–20% of pediatric CKD patients showed protein-energy wasting [170]. Recently, an RCT was conducted with 48 CKD patients including 33 undergoing hemodialysis to compare the effects of two different doses of zinc supplementation (15 and 30 mg/day) given for 12 months [153]. There was no significant change in mean serum zinc level in children in either group. On the other hand, a small but positive and significant change in body mass as well as normalization of body mass index (BMI) *Z*-score, hypoalbuminemia, hypozincemia, and high CRP was noted, especially with a dose of 30 mg/day, which suggested that zinc supplementation could be beneficial for nutritional status in children with CKD. Another interventional study of 40 hemodialysis patients aged between 5 and 18 years old and given daily zinc supplementation of 50–100 mg for 90 days found that serum zinc was significantly increased from 53.2 ± 8.15 to 90.75 ± 12.2 μg/dL (*p* = 0.001) [157].

## 13. Optimal Serum Zinc Level

The optimal level of zinc in serum remains controversial. The American Society for Parenteral and Enteral Nutrition guidelines suggests that trace minerals, including zinc, should be provided to critically ill patients [171]. Additionally, according to the European Society for Clinical Nutrition and Metabolism guidelines, zinc levels should be measured as a part of nutrition screening [172]. The Japanese Practical Guidelines have generally proposed a level > 80 μg/dL as normal zinc status in all individuals [29]. However, a recent nationally representative cross-sectional study that enrolled subjects from participants in the National Health and Nutrition Examination Survey [173] showed that a higher zinc level per every 10 μg/dL was associated with a 1.12-times higher risk for diabetes mellitus and a 1.23-times higher risk for CVD in those with a serum zinc level ≥ 100 μg/dL. Furthermore, each 10 μg/dL increase was also associated with a 1.40-fold increase in stroke in participants with a serum zinc level ≥120 μg/dL [173]. The mechanisms underlying these relationships are unclear, though it may be important to avoid hyperzincemia in regard to CVD events in the general population. Regarding hemodialysis patients, a recent study recommended a lower level of serum zinc (78.3 μg/dL) because of the potential for copper deficiency [174]. Additional studies are needed to determine the optimal zinc blood level in the general population, as well as in pre-dialysis CKD patients and those receiving hemodialysis treatments.

## 14. Conclusions

In patients with CKD, zinc deficiency is common and shown by taste change, decreased food intake, and/or increased urinary excretion. Zinc deficiency is also known to be associated with various risk factors for CVD, including increased blood pressure, dyslipidemia, type 2 diabetes mellitus, inflammation, and oxidative stress. Various clinical studies have revealed that zinc intake/supplementation can increase blood zinc levels in patients with CKD. In addition, zinc may prevent phosphate-induced arterial calcification by inducing the production of TNFIAP3 as well as suppressing activation of NF-kB. High-quality prospective cohort studies and RCTs are needed to provide evidence for zinc intake/supplementation as an effective therapeutic tool for preventing CVD events in patients with CKD, including those undergoing hemodialysis.

## Figures and Tables

**Figure 1 nutrients-13-01680-f001:**
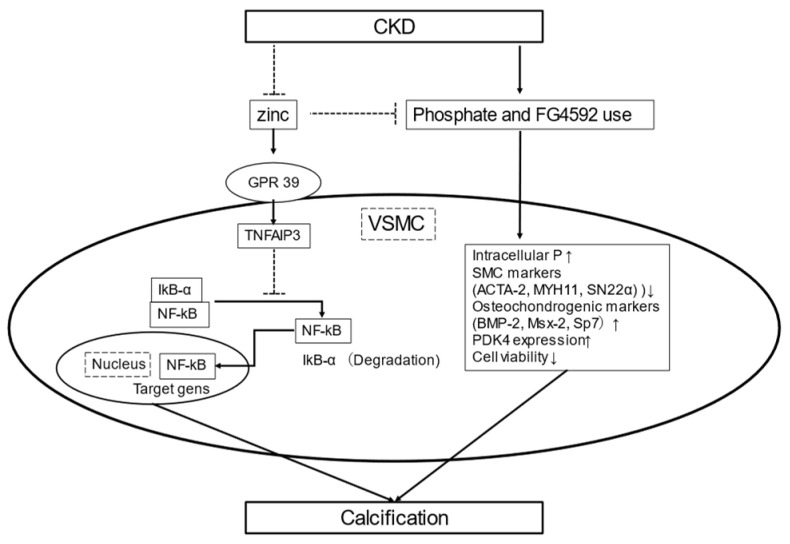
Schematic illustration of zinc and calcification. CKD induces hypozincemia and hyperphosphatemia. Zinc supplementation may increase zinc finger protein TNFAIP3 levels by upregulating zinc-sensing receptor ZnR/GPR39-dependent TNFAIP3 gene expression. Increased TNFAIP3 inhibits NF-kB activation and osteo-/chondrogenic reprograming, resulting in suppression of phosphate-induced VSMC calcification [17]. FG4592, an orally bioavailable PHI, promotes phosphate uptake in VSMCs and phosphate-induced loss of smooth muscle cell markers (ACTA-2, MYH11, SM22a) and also enhances osteochondrogenic gene expression (Msx-2, BMP-2, Sp7). Zinc inhibits FG4592-aggravated calcification caused by high phosphate by maintaining the VSMC phenotype, decreasing phosphate uptake, and lowering osteochondrogenic gene expression and levels of PDK4, as well as preserving Runx2 phosphorylation and cell variability [16]. Abbreviations: ACTA-2, smooth muscle a-2 actin; BMP-2, bone morphogenic protein-2; CKD, chronic kidney disease; NF-kB, nuclear factor kappa light chain enhancer of activated B; Msx-2, Msh Homeobox 2; MYH11, smooth muscle myosin heavy chain 11; PDK4, pyruvate dehydrogenase kinase 4, PHI, prolyl hydroxylase inhibitors; Runx2, runt-related transcription factor 2; TNFAIP3, TNFa-induced protein 3; VSMCs; vasculature smooth muscle cells.

**Figure 2 nutrients-13-01680-f002:**
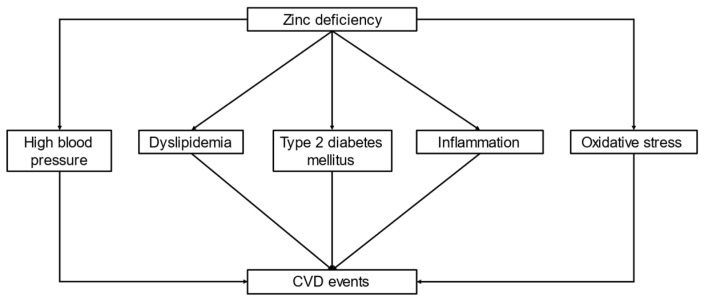
Association of zinc deficiency. Zinc deficiency is associated with major risk factors for CVD, including higher blood pressure, dyslipidemia, type 2 diabetes mellitus, inflammation, and oxidative stress. Zinc deficiency is associated with CVD events in CKD patients, including those undergoing hemodialysis. Abbreviations: CKD, chronic kidney disease; CVD, cardio vascular disease.

**Table 1 nutrients-13-01680-t001:** Summary of observational studies regarding zinc deficiency in patients with CKD.

Author, Year [30]	Country	Number of CKD/HD Patients	Number of Healthy Subjects	Sample	Zinc level, CKD vs. Control †
CKD					
Tavares et al. 2020 [40]	Brazil	21	22	Plasma	70.1 ± 19.2 vs. 123.2 ± 24.6 (μg/dL)
Shen et al. 2020 [41]	China	193	173	Plasma	188 vs. 229 (μg/dL)
Damianaki et al. 2020 [42]	Switzerland	108	42	Plasma	60.6 ± 10.6 vs. 66.4 ± 10.1 (μg/dL)
Pan et al. 2019 [43]	Taiwan	204	2853	Serum	76.9 ± 1.29 vs. 82.8 ± 0.67 (μg/dL)
Aziz et al. 2016 [44]	Iraq	49	42	Plasma	83 ± 10 vs. 112 ± 19 (μg/dL)
Mafra et al. 2002 [45]	Brazil	29	19	Plasma	74 ± 17.7 vs. 82.1 ± 15.5 (μg/dL)
HD					
Hasanato 2014 [46]	Saudi Arabia	42	18	Plasma	9.5 vs. 13.2 (nmol/L)
Lobo et al. 2013 [47]	Brazil	45	20	Plasma	54.9 ± 16.1 vs. 78.8 ± 9.4 (μg/dL)
Guo et al. 2011 [48]	Taiwan	20	20	Plasma	68 ± 3 vs. 76 ± 8 (μg/L)
Dashti-Khavidaki et al. 2010 [49]	Iran	94	47	Serum	69.2 ± 17.3 vs. 82.9 ± 14.8 (μg/dL)
Kiziltas et al. 2008 [50]	Turkey	30	30	Serum	15.7 ± 1.25 vs. 21.2 ± 1.44 (μmol/L)
Batista et al. 2006 [51]	Brazil	30	20	Plasma	81.2 ± 19.8 vs. 93.3 ± 12.1 (μg/dL)

† Zinc level values are shown as the mean ± standard deviation. Abbreviations: CKD, chronic kidney disease; HD, hemodialysis.

**Table 2 nutrients-13-01680-t002:** Summary of cohort studies regarding blood zinc levels and CVD mortality.

Author, Year, (Reference)	Country	Number of Subjects	Age (Years) †	Follow-Up Period (years) ‡	Number of CVD Deaths	Association of Lower Blood Zinc Levels with Higher CVD Mortality
Bates et al. 2011 [25]	UK	1054 (general population)	≥65 years old Male: 75.8 ± 6.9 Female: 77.3 ± 7.9	n/a	189	Yes (HR 0.79; 95% CI 0.72–0.87)
Pilz et al. 2009 [140]	Germany	3316 (patients referred for coronary angiopathy)	Male: 62 ± 11 Female: 65 ± 10	7.75	484	Yes (HR 1.10; 95% CI 1.01–1.21) (Reference: high serum zinc group)
Leone et al. 2006 [142]	France	4035 males (general population)	30–60 years old 43 ± 5 (alive) 44 ± 4 (dead)	18 ± 2.9	56	No (RR 0.7; 95% CI 0.3–1.5)
Marniemi et al. 1998 [143]	Finland	344 (general population)	≥65 years old 65–69 (*n* = 99), 70–74 (*n* = 98) 75–80 (*n* = 84), ≥80−(*n* = 63)	13	142	No (HR 0.77; 95% CI 0.42–1.41)

† Age shown as mean ± standard deviation or range (lower limit, upper limit). ‡ Follow-up period shown as mean or mean ± standard deviation. Abbreviations: CVD, cardiovascular disease; HR, hazard ratio; NA, not available; RR, relatively risk.

**Table 3 nutrients-13-01680-t003:** Summary of observational studies regarding zinc intake and CVD mortality.

Author, Year (Reference)	Country	Number of Subjects	Age (Years) †	Follow-Up Period (years)	Number of CVD Deaths	Outcomes
Chen et al. 2019 [146]	USA	30,899	46.9	6.1	945	Adequate nutrient intake of zinc associated with lower CVD mortality (RR = 0.50; 95% CI 0.36–0.71).
Shi et al. 2018 [147]	China	2832	47.1	9.8	70	Dietary zinc intake not related to CVD mortality.
Eshak et al. 2018 [26]	Japan	58,646	40–79	19.3	3388	Higher intake of zinc inversely associated with mortality from coronary heart disease (*n* = 702) in males; 0.68 (0.58–1.03; *p*-trend = 0.05) but not females; 1.13 (0.71–1.49; *p*-trend = 0.61).
Bates et al. 2011 [25]	UK	1054	75.8 ± 6.9 (males) 77.3 ± 7.9 (females)	n/a	189	Plasma zinc associated with vascular disease mortality (HR 0.73; 95% CI 0.61–0.88).
Lee et al. 2005 [148]	USA	34,492	(55–69)	>15	1767	Inverse association of dietary zinc with CVD mortality.

† Age shown as mean ± standard deviation or range (lower limit, upper limit). ‡ Follow-up period shown as mean. Abbreviations: CVD: cardiovascular disease; NA, not available; RR: relatively risk.

**Table 4 nutrients-13-01680-t004:** Summary of RCTs of zinc supplementation in patients with hemodialysis.

Author, Year (Reference)	Country	Number of Subjects	Age (Years) †	Elemental Zinc Dose (mg/day)	Administration Duration (Days)	Outcomes
Escobedo-Monge et al. 2019 [153]	Peru	48 (children)	12.8 ± 4	15/30	365	Increase: BMI (30 mg/day group only)
Kobayashi et al. 2015 [156]	Japan	70	69 ± 10	34	90/180/270/360	Increase: serum zinc Decrease: serum copper, ferritin
El-Shazly et al. 2015 [157]	Egypt	30	13.2 ± 2.1	16.5	90	Increase: serum zinc, BMI Decrease: serum leptin
Tonelli et al. 2015 [154]	Canada	150	62	25 and 50	90 and 180	None
Argani et al. 2014 [114]	Iran	60	(50,60)	90	60	Increase: serum zinc, albumin, hemoglobin, BMI Decrease: serum leptin
Pakfetrat et al. 2013 [158]	Iran	97	51.6 ± 16.8	50	43	Increase: serum zinc Decrease: homocysteine
Mazani et al. 2013 [159]	Iran	65	52.7 ± 12.6	100	60	Increase: serum zinc, GSH, MDA, SOD, TAC
Guo and Wang. 2013 [160]	Taiwan	65	59.7 ± 9.2	11	56	Increase: plasma zinc, albumin, hemoglobin, hematocrit, nPNA, SOD, vitamin C, vitamin E, CD4, D19 Decrease: plasma copper, CRP, MDA INF-b, TNF-𝛼,
Rahimi-Ardabili et al. 2012 [161]	Iran	60	52.7 ± 12.7	100	60	Increase: Apo-AI, HDL-C, PON
Roozbeh et al. 2009 [115]	Iran	53	55.7	45	42	Increase: serum zinc, TC, HDL-C, LDL-C, TG
Rashidi et al. 2009 [162]	Iran	55	57.6	45	42	Increase: serum zinc
Nava-Hernandez and Amato 2005 [163]	Mexico	25	16.6	100	90	n/a
Matson et al. 2003 [164]	UK	15	60 (31–76)	45	42	Not significant
Chevalier et al. 2002 [165]	USA	27	51.9	50	40/90/90	Increase: serum zinc, LDL-C
Candan et al. 2002 [166]	Turkey	34	45.6 (28,64)	20	90	Increase: serum zinc Decrease: lipid peroxidation osmotic fragility
Jern et al. 2000 [167]	USA	14	56.5 (23,80)	45	40/90	Increase: serum zinc, nPNA
Brodersen et al. 1995 [168]	Germany	40	60	60	112	Increase: serum zinc

Note. † Age is shown as mean, mean ± standard deviation, or mean (lower limit, upper limit). Abbreviations: Apo-AI, apolipoprotein AI; BMI, body mass index; Ccr, creatinine clearance rate; CRP, C-reactive protein; ESA, erythropoiesis-stimulating agent; ERI, ESA resistance index; GFR, glomerular filtration rate; GSH, whole blood glutathione peroxidase; HDL-C, high-density lipoprotein cholesterol; IL, interleukin; LDL-C, low-density lipoprotein cholesterol; MDA, malondialdehyde; NA, not available; nPNA, normalized protein equivalent of nitrogen appearance; PON, paraoxonase; SOD, superoxide dismutase; TAC, total antioxidant capacity; TC, total cholesterol; TG, triglyceride; TNF, tumor necrosis factor.

## Data Availability

Not applicable.

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
