# Peer review of "Association of Zinc Deficiency with Development of CVD Events in Patients with CKD"

_nutrients, 2021, doi:10.3390/nu13051680_

Round 1

Reviewer 1 Report

1.This paper reviewed zinc deficiency and its effects on CVD and CKD patients. The authors reviewed most of the related papers, and is clearly written. They provided good information between zinc deficiency and the two diseases (CVD and CKD).

2. It is suggested to reorganize all the paper. 

a. This review has 2 major parts: zinc deficiency and CVD, and zinc deficiency and CKD. In fact, zinc deficiency and CVD is the major theme which includes 7 sections: 4, 5, 6, 7, 8,10, and 11 among the total 13 sections. Zinc deficiency and CKD only described in Sections 3, 9,12, and 13.

b. There are 2 figures in this review, both of them emphasize on zinc and CVD. Fig.1 shows zinc and calcification, and Fig. 2 shows zinc deficiency and CVD events.

c. The contents of this review is more likely: Effects of zinc deficiency on CVD and CKD, or 'Association of zinc deficiency with development of CVD events in patients with CKD' (as they mentioned in the Introduction, Lines 65-66).

d. They reviewed zinc and CVD from Section 4 to 8, and Section 10 to 11. It is better to remove Section 9 (zinc and progression of CKD) after Section 11. Then you could put all the CVD sections together.

3. In p.3, Lines 98-102, the factors related to low zinc level in CKD patients were outlined. This is based on the References of 36 and 37. Most  CKD patients have lower plasma zinc levels were identified as early as 1950 and 1970, and have been reported continuously. The cause of the low plasma zinc levels in CKD patients has been explained by several factors as mentioned by the Reference 36 and 37. But in a recent new report: (In anemia zinc is recruited from bone and plasma to produce new red blood cells, Chen et al., J. Inorganic Biochemistry 210 (2020)111172), the authors indicate that the lower plasma zinc level in animals might be the redistribution of zinc throughout the animal body, besides the above factors. You may like to discuss it or add to your manuscript. 

4. Minor mistake: line 143: a-macroglobulin--alpha-macroglobulin.

Author Response

Reviewer 1

Comments and Suggestions for Authors

Comment 1

This paper reviewed zinc deficiency and its effects on CVD and CKD patients. The authors reviewed most of the related papers, and is clearly written. They provided good information between zinc deficiency and the two diseases (CVD and CKD).

We appreciate the kind consideration of our study and helpful comments from the reviewer.

Comment 2

It is suggested to reorganize all the paper.

  1. This review has 2 major parts: zinc deficiency and CVD, and zinc deficiency and CKD. In fact, zinc deficiency and CVD is the major theme which includes 7 sections: 4, 5, 6, 7, 8,10, and 11 among the total 13 sections. Zinc deficiency and CKD only described in Sections 3, 9,12, and 13.
  2. There are 2 figures in this review, both of them emphasize on zinc and CVD. Fig.1 shows zinc and calcification, and Fig. 2 shows zinc deficiency and CVD events.
  3. The contents of this review is more likely: Effects of zinc deficiency on CVD and CKD, or 'Association of zinc deficiency with development of CVD events in patients with CKD' (as they mentioned in the Introduction, Lines 65-66).
  4. They reviewed zinc and CVD from Section 4 to 8, and Section 10 to 11. It is better to remove Section 9 (zinc and progression of CKD) after Section 11. Then you could put all the CVD sections together.

We appreciate these constructive suggestions, which were very helpful for improving the manuscript.

As mentioned by the reviewer, this manuscript has two major parts; zinc deficiency and CVD. However, evidence related to zinc deficiency and progression of CKD is not as plentiful as that regarding zinc deficiency and CVD events. As a result, the amount of text as well as numbers of figures, tables, and sections are not balanced between those two topics. In the revised version, we have changed the title of the manuscript to “Association of zinc deficiency with development of CVD events in patients with CKD”, as suggested by the reviewer. In addition, Section 9 has been moved and now appears after former Section 11 (order of sections in revised version: 9. Zinc levels and CVD events, 10. Zinc and CVD mortality, 11. Zinc and progression of CKD).

Comment 3

In p.3, Lines 98-102, the factors related to low zinc level in CKD patients were outlined. This is based on the References of 36 and 37. Most CKD patients have lower plasma zinc levels were identified as early as 1950 and 1970, and have been reported continuously. The cause of the low plasma zinc levels in CKD patients has been explained by several factors as mentioned by the Reference 36 and 37. But in a recent new report: (In anemia zinc is recruited from bone and plasma to produce new red blood cells, Chen et al., J. Inorganic Biochemistry 210 (2020)111172), the authors indicate that the lower plasma zinc level in animals might be the redistribution of zinc throughout the animal body, besides the above factors. You may like to discuss it or add to your manuscript.

We appreciate the reviewer noting this novel report (Chen et al., J. Inorganic Biochemistry 210 (2020)111172). Redistribution of zinc from plasma and bone to bone marrow must be an important mechanism related to zinc deficiency in CKD and acute anemia cases. This study is now cited in the revised manuscript along with related discussion (page 3, lines 107-112 of revised version).

Comment 4

Minor mistake: line 143: a-macroglobulin--alpha-macroglobulin.

We apologize for the mistake. That has been corrected to “α-macroglobulin”(page 4, line 152 of revised version).

Reviewer 2 Report

This is a well-researched, comprehensive, and generally well-written review on the effects of Zn deficiency, which is a topic of interest. I think that they bring the current literature together well in this paper. I have only some minor points to make:

1. The wording and grammar of some sentences are incorrect or poor and these need to be re-written. Examples are:

Some studies have demonstrated that a negative zinc balance in a CKD patient may be due to decreased intestinal absorption, decreased food intake, uremic toxicity, bioavailability, and/or increased loss, such as through the face, urine, or dialysis [36, 37].

In undergoing hemodialysis patients who show chronic malnutrition along with low serum albumin, this should be considered as another confounding factor to interpret plasma zinc concentration, because albumin is the primary carrier protein for circulating zinc [57].

Physicians should give careful attention to protein restriction and the use of phosphate binder medication in respect to zinc deficiency in the clinical setting of CKD management.

Clinical trials are needed to examine whether zinc supplementation attenuates aortic calcification and CVD events in CKD patients with high phosphate overload or treated with PHI are anticipated

2. Figure 2 should not be labelled ‘Consequences of Zinc deficiency’. There are multiple causes of HT and hyperlipidaemia, inflammation and oxidative stress, and Zn deficiency may contribute to but is not a sole cause of these. It could be labelled ‘Associations of Zinc deficiency’, (which is what the authors do indicate immediately after this heading).

Author Response

Reviewer 2

This is a well-researched, comprehensive, and generally well-written review on the effects of Zn deficiency, which is a topic of interest. I think that they bring the current literature together well in this paper. I have only some minor points to make:

We appreciate the helpful comments by the reviewer and suggestions to improve our manuscript.

Comment 1

The wording and grammar of some sentences are incorrect or poor and these need to be re-written. Examples are:

Some studies have demonstrated that a negative zinc balance in a CKD patient may be due to decreased intestinal absorption, decreased food intake, uremic toxicity, bioavailability, and/or increased loss, such as through the face, urine, or dialysis [36, 37].

In undergoing hemodialysis patients who show chronic malnutrition along with low serum albumin, this should be considered as another confounding factor to interpret plasma zinc concentration, because albumin is the primary carrier protein for circulating zinc [57].

Physicians should give careful attention to protein restriction and the use of phosphate binder medication in respect to zinc deficiency in the clinical setting of CKD management.

Clinical trials are needed to examine whether zinc supplementation attenuates aortic calcification and CVD events in CKD patients with high phosphate overload or treated with PHI are anticipated

We appreciate the reviewer pointing out the poor wording and grammar in some of the sentences. Those have been corrected in the revised version and the text has been carefully checked by a native English speaker. (page 3, lines 103-105; page 4, lines 156-160; page 4, lines 170-173; and page 6, lines 243-245, respectively, in the revised version).

Comment 2

Figure 2 should not be labelled ‘Consequences of Zinc deficiency’. There are multiple causes of HT and hyperlipidaemia, inflammation and oxidative stress, and Zn deficiency may contribute to but is not a sole cause of these. It could be labelled ‘Associations of Zinc deficiency’, (which is what the authors do indicate immediately after this heading).

We agree with the reviewer’s suggestion and have changed the title of Figure 2 to “Associations of zinc deficiency”.

Round 2

Reviewer 1 Report

The paper is well reorganized.

Only one suggestion:

Lines 704-707, the legend of Figure 2 may be simplified as: "Zinc deficiency and risk factors for CVD". The figure describles the contens clearly, it seems no necessary to repeat it in the  legend.

One small miss:. line 399; with,  correct to 'with'.